


# Quantifying the extremeness of precipitation across scales

Paul Voit[1] and Maik Heistermann[1]

[1]Institute for Environmental Sciences and Geography, University of Potsdam, Potsdam, Germany

**Correspondence:** Paul Voit (voit@uni-potsdam.de)

**Abstract.**

Quantifying the extremeness of a heavy precipitation event is important to compare different events, to analyze trends in frequency and amplitude, and to understand related impacts on the ground. While such impacts depend on the event's spatial extent and duration, many indices neglect at least one of these aspects. In 2014, however, Müller and Kaspar suggested, in this

journal, the weather extremity index (*WEI*) which quantifies not only the extremeness of an event, but identifies the spatial and temporal scale at which the event was most extreme. While the *WEI* is informative, it does not account for the fact that an event can be extreme at various spatial and temporal scales. Such an event could trigger - simultaneously or subsequently - different kinds of processes and related impacts, such as flash floods and large-scale fluvial floods, which can overlay and amplify each other, so that they essentially become compound events. To better understand and detect the compound nature of precipitation

events, we suggest to complement the original *WEI*, and refer to this complement as the "cross-scale weather extremity index" (*xWEI*). Unlike the original *WEI* index, *xWEI* does not aim to detect the spatio-temporal scale of maximum extremeness, but to integrate extremeness over relevant scales.

Based on a set of 101 extreme precipitation events in Germany, we outline and demonstrate the computation of both indices, *WEI* and *xWEI*, and analyse how the choice of an index affects the rating and ranking of these events. To that end, we use

hourly radar-based precipitation estimates for all of Germany at a spatial resolution of 1 x 1 km, available since 2001. We find that the choice of the index can lead to considerable differences in the assessment of past events, but that the most extreme events are ranked consistently, independently of the index. Even for these cases, though, the *xWEI* index can reveal cross-scale properties which would otherwise remain hidden. Among the analysed events was also the disastrous precipitation event from July 2021 which devastated large parts of western Germany. This event outranks all other analysed events by far - both with

regard to *WEI* and *xWEI*.

While demonstrating the added value of the cross-scale index, we also identify various methodological challenges along the required computational workflow: these include the parameter estimation for the extreme value distributions, the definition of maximum spatial extent and temporal duration, as well as the weighting of extremeness at different scales. These challenges, however, also represent opportunities to adjust the retrieval of *WEI* and *xWEI* to specific user requirements and application

scenarios. We conclude that the proposed cross-scale extremity index can provide substantial complementary information to existing indices, and could hence be a valuable instrument in both disaster risk management and research.





# 1 Introduction

Quantifying heavy precipitation events (HPEs) is important as these events can have significant impacts on nature and society (Lengfeld et al., 2020). The devastating flood following the event in July 2021 in western Germany is a recent example. Extreme

precipitation can cause different flood types (flash-, pluvial and fluvial floods), erosion as well as landslides (Leonarduzzi et al., 2021; Ozturk et al., 2018; Zêzere et al., 2005). While HPEs are already among the costliest natural disasters in Europe (Gvoždíková et al., 2019), climate change conditions are expected to lead to an increase in frequency and intensity of HPEs (Christensen and Christensen, 2003; Field et al., 2012; Pryor et al., 2014). Warmer and wetter conditions could additionally impact the spatial extent of precipitation features, which might lead to rain cells capable of producing up to almost 20 % more

rain per degree warming (Lochbihler et al., 2019), whereas Prein et al. (2017), state an expected increase of precipitation intensities of about 7 % per degree warming. Despite the different numbers, this is suggesting a significant increase in HPEs and the connected impacts (Zhang et al., 2019). Only an enhanced understanding regarding severity, duration and frequency of HPEs will enable us to adapt to these events by appropriate hazard mitigation and management strategies.

Impacts following HPEs are manyfold and caused by different mechanisms. Short duration rainfall with high intensities is

associated with flash- or pluvial floods due to saturation and infiltration excess while persistent precipitation episodes on the daily scale can lead to large-scale fluvial floods (Ramos et al., 2017). As one HPE can be extreme on different spatio-temporal scales simultaneously, it can trigger different types of impacts which can overlay each other. Impacts from extreme weather events can be caused by a single variable being extreme or an accumulation of not necessarily extreme variables (Liu et al., 2018). The latter is also referred to as compound events which the IPCC (Seneviratne et al., 2012) defined as:

1. two or more extreme events occurring simultaneously or successively

   2. combinations of extreme events with underlying conditions that amplify the impact of these events

   3. combinations of events that are not themselves extreme but lead to an extreme event or impact when combined.

Thieken et al. (2022) adopted this concept and described some of the most destructive floods that have been observed in Germany as compound inland floods, as these were a chain of interacting and cascading events. In August 2002, for example,

the city of Dresden was hit by consecutive flood events which were effectively triggered by the same rainfall event. First, pluvial flooding occurred due to high intensity rainfalls with short duration (August 12, 2002). The following day the city was hit by a flash flood originating from the small rivers Weißeritz and Lockwitzbach. This was followed on August 17 by a flood wave of the river Elbe (fluvial flooding). Further downstream, this led to dike breaches and caused huge inundations of the hinterland (DKKV, 2003; Thieken et al., 2022). This flood from 2002 is just one example of how one rainfall event can

be extreme - and hence impactful - on various spatio-temporal scales. The event contained high intensity episodes that were extreme at short durations (hours) while the cumulative event depth was extreme at a long duration (days), too. Rainfall at long durations increases the soil moisture content which can in turn amplify the impacts of pluvial and flash floods caused by extreme precipitation at shorter durations (Schröter et al., 2015). The compound nature of impacts following an HPE is therefore often caused by the compound nature of the precipitation event itself.





A precipitation event is a substantial precipitation activity that displays a certain level of temporal and spatial coherence. Traditionally, the definition of precipitation extremeness is based on the occurrence probability (or return period) at a specific point (e.g. a rain gauge) and a specific duration. However, point-based measures do not account for the area affected by extreme precipitation, which is a fundamental property: hydrologically, the affected area controls the scale at which runoff can concentrate within a network of streams and rivers which again influences the type of impact. At the same time, and more

intuitively, it describes the area in which certain local impacts such as pluvial flooding can occur. We often assume, implicitly, that high-intensity rainfall at short durations affects small areas while extreme rainfall at long durations comes with large affected areas (Lengfeld et al., 2021a; Orlanski, 1975) This implicit assumption might hide, however, fundamental properties that could define the impact-relevance of an event, and could be replaced by explicitly accounting for the affected area at various durations.

Müller and Kaspar (2014) addressed exactly that gap. They quantified, for a fixed spatial domain and a fixed time window, the extremeness of an event at different spatial and temporal scales, and suggested the "weather extremity index" (*WEI*) that corresponds to the maximum value of extremeness over all considered spatial and temporal scales. The *WEI* was used for detecting and ranking HPEs (Gvoždíková et al., 2019; Minářová et al., 2018) and was adopted by the German Weather Service (DWD) to evaluate HPEs for the event catalogue CatRaRE (CATalogue of Radar-based heavy Rainfall Events, (Lengfeld et al.,

2021a)). Another approach that takes into account the spatio-temporal extremeness of precipitation events was suggested by Ramos et al. (2017). In their study they considered the affected area by accumulating grid cells with precipitation anomalies over each time scale and ranked past HPEs for each duration for the Iberian Peninsula (Ramos et al., 2017). A similar study was conducted for the Indian Western Himalayas (Raj et al., 2021). Looking at different durations independently, Ramos et al. (2017) and Raj et al. (2021) observed that the same event can be extreme at different durations which could be considered a

property of a compound event. Reducing the extremeness of one event to only the scale of maximum extremeness could hence conceal how extremeness extends across temporal or spatial scales, or, in other words, to what degree the event was "extreme across scales". Concentrating only on the duration and extent at which an HPE showed its maximum extremeness might be valid for some applications contexts but for others, it is important to quantify how much the extremeness extended across scales. This might not only apply to the causation of impacts such as floods and landslides, but also to the adequate disaster

response: a severe local impact attracts disaster response resources from a certain radius, depending on severity. If an event is extreme across scales, these radii might overlap in a way that multiple local events draw away required resources from each other.

    In this study, we therefore suggest a simple, but important extension (or complement) of the *WEI* suggested by Müller and Kaspar (2014), and we demonstrate that this "cross-scale index" is able to shed new light on the compound properties of

extreme precipitation. To that end, we will compare the original *WEI* to the proposed cross-scale index for a set of 100 precipitation events selected from the CatRaRE event catalogue published by the Germany Weather Service (Deutscher Wetterdienst, DWD) (Lengfeld et al., 2021a). In addition, we analyzed the event in July 2021 in West Germany in order to put its extremeness into context.





The analysis is based on a gauge-adjusted, radar-based precipitation product which provides twenty years of quality-
controlled hourly rainfall depths (from 2001 to 2020) on a one kilometer grid across Germany (the RADKLIM dataset,
Winterrath et al. (2018)). Hence, this study also addresses the methodological challenges and opportunities that arise from
the use of such a data set with regard to the estimation of extreme value distributions at individual grid points. More specifi-
cally, the use of the RADKLIM dataset constitutes an inherent trade-off: as opposed to sparse rain gauge data, it provides high
spatial resolution, coverage and representativeness; yet, the length of the time series (20 years) introduces uncertainties as to
the estimation of precipitation levels at long return periods. We hence explore options for a robust estimation of GEV (general-
ized extreme value distribution)-parameters on a per-pixel basis, including the Region-of-Interest method (Burn, 1990) and the
duration dependent GEV parameter estimation (Koutsoyiannis et al., 1998; Ulrich et al., 2020; Fauer et al., 2021). We would
like to emphasize, though, that the present study is about the concept of a cross-scale extremity index; spatially distributed
values of return periods are required to obtain the index, but other sources or methods to obtain the required return periods.
In section 2 of this paper, we will briefly introduce the two datasets, RADKLIM and CatRaRE. Section 3 will outline the
methodological details, including the estimation of GEV-parameters and the computation of the original *WEI* index and the
suggested cross-scale extension. Section 4 presents the results of our analysis: we demonstrate the effect of the proposed index
on the ranking of precipitation events with regard to extremeness, and highlight the properties of the new index for two case
studies.

## 2  Data

### 2.1  Precipitation Data

For this study, we use the RADKLIM_RW_2017.002 dataset by the DWD (Winterrath et al., 2018). Since 2001, the DWD
has been operating a network of C-band weather radars (17 radars as of today). The product chain to the hourly precipitation
estimate is referred to as RADOLAN (RADar OnLine ANeichung, Winterrath et al. (2012)), and includes comprehensive
steps of quality control and corrections, including the final step of adjustment by rain gauges. For the RADKLIM product,
the data from 2001 to 2020 was consistently reanalysed using state-of-the-art algorithms as well as an extended set of rain
gauge observations for the adjustment step. It was shown that this procedure minimizes the occurrence of artifacts (Lengfeld
et al., 2019) and provides a promising dataset for climatological applications (Pöschmann et al., 2021). The resulting dataset
is a Germany-wide precipitation field of hourly precipitation sums at an extent of 1100 x 900 km and at a resolution of 1 x
1 km and is available on the DWD open data server (Winterrath et al., 2018). As the event in July 2021 was not yet included
in the latest RADKLIM reanalysis, we used the operational RADOLAN product instead for this one event (DWD, 2021). The
hourly dataset was accumulated to a set of durations. We chose commonly used duration levels, most of which are also used
by the DWD, that represent intense precipitation with short durations as well as moderate to intense long-lasting precipitation
episodes (Fauer et al., 2021; Lengfeld et al., 2021a):





**Table 1.** Events used for case studies. The short name was constructed from an acronym that specifies the region in which the event occurred (mostly the federal state), the month and the year.

| Short name | Region | CatRaRE ID | Start | End |
|---|---|---|---|---|
| BE/Jun2017 | Berlin | 17695 | 29 Jun 2017, 10:50 | 30 Jun 2017, 10:50 |
| BW/May2016 | Baden Württemberg | 16058 | 29 May 2016, 12:50 | 30 May 2016, 06:50 |
| NI/Jul2002 | Lower Saxony | 1239 | 17 Jul 2002, 01:50 | 19 Jul 2002 01:50 |
| NI/Jul2017 | Lower Saxony | 17961 | 24 Jul 2017, 07:50 | 26 Jul 2017, 07:50 |
| NW/Jul2014 | North Rhine-Westphalia | 14213 | 28 July 2014, 13:50 | 28 Jul 2014, 22:50 |
| SL/May2018 | Saarland | 19168 | 30 May 2018, 19:50 | 01 Jun 2018, 19:50 |
| SN/Aug2002 | Saxony | 1798 | 12 Aug 2002, 02:50 | 13 Aug 2002, 02:50 |
| SN/May2018 | Saxony | 12316 | 30 May 2013, 15:50 | 02 Jun 2013, 15:50 |
| WG/Jul2021 | West-Germany | - | 12 Jul 2021, 00:50 | 17 Jul 2021, 00:50 |

$$d \in \{01\,\text{h}, 02\,\text{h}, 04\,\text{h}, 06\,\text{h}, 12\,\text{h}, 24\,\text{h}, 48\,\text{h}, 72\,\text{h}\} \quad d = duration \tag{1}$$

## 2.2 CatRaRE

The DWD extracted more than 20.000 HPEs with durations between 1 h to 72 h from 20 years of radar data (RADKLIM) in Germany. Each HPE is listed with parameters such as date, time, duration, mean and maximum precipitation, severity indices as well as geographical and demographical information. There are two versions of the catalog that use different thresh-
olds (exceedance of the warning level three for severe precipitation by the DWD and the exceedance of a return period of 5 years). The catalog is updated every year and is openly available (Lengfeld et al., 2021b). The WEI was used to determine the most extreme duration level of an event and is one of the attributes listed in CatRaRE. For this study, we used the CatRaRE version that is based on the exceedance of warning level three (25 mm in one hour or 35 mm in six hours, CatRaRE_2001_2020_W3_Eta_v2021_01) as an objective basis to select the 100 most extreme precipitation events in Ger-
many between 2001 and 2020. Nine HPEs will be specifically discussed in this study, and are hence detailed in Tab. 1.

## 3 Methods

In this study we evaluate HPEs based on the WEI by Müller and Kaspar (2014), and extend this index to represent extremeness across spatio-temporal scales. We will refer to this as the cross-scale weather extremity index *xWEI*. As both indices are based





on the calculation of return periods and therefore rely on extreme value statistics, we will first describe the three different
methods we used to derive the parameters for the GEV distribution, and then outline the calculation of *WEI* and *xWEI*. The
process is as follows:

1. Calculate the parameters of the GEV distribution for each pixel in the RADKLIM_RW_2017.002 dataset with different
   methods (cellwise GEV, Region-of-interest, duration dependent GEV distribution).

2. Evaluate 101 selected HPEs using the *WEI* and *xWEI*, based on each method of GEV-parameter estimation.

3. Selection of events for case studies based on ranking results.

## 3.1 Calculation of return periods

*WEI* and *xWEI* require return periods for each pixel and each duration in the spatial domain. We estimated return periods
for each pixel of the RADOLAN grid using the GEV distribution which was found to be suitable for modeling precipitation
extremes (Fowler and Kilsby, 2003), and which was used in previous studies that applied the *WEI* (Gvoždíková et al., 2019;
Minářová et al., 2018; Müller and Kaspar, 2014). While Müller and Kaspar (2014) proposed an interpolation of return periods
derived from station data to a grid, we instead use gridded precipitation data and perform cellwise extreme value statistics
to derive return periods. This way we avoid the uncertain interpolation; yet, the time series used for estimating the GEV
parameters are comparatively short (20 years). To address this issue, we compared different methods that help improving the
robustness of the parameter estimation, all of which are based on the annual maximum values for each duration and each grid
cell:

1. As a reference method, we fitted the GEV distribution to the series of annual maxima for each cell and each duration
   with the R-package "extRemes" (Gilleland and Katz, 2016).

2. Region-of-interest (ROI): We included information from neighboring cells to make the GEV parameter estimation more
   robust towards small-scale variability in the RADKLIM_RW_2017.002 dataset. The series of annual maxima from the
   pixels in a 19 x 19 km box around the pixel of interest are weighted by distance to the center and are included in the
   estimation of the GEV parameters for this pixel. This method, described in detail in Burn (1990), was also used by
   Müller and Kaspar (2014).

3. Alternatively, we took advantage of the parameter dependence between different durations in order to make the parameter
   estimation more robust (Koutsoyiannis et al., 1998). While the reference approach fits the GEV parameters independently
   for each duration, this approach introduces a duration dependent scale and location parameter in the GEV distribution
   which is then estimated simultaneously for all durations. This approach is considered consistent because it prevents the
   crossing of quantiles (Fauer et al., 2021), but it also is computationally more efficient. To fit the parameters for this
   duration dependent GEV (dGEV) distribution for each pixel we used the R-package "IDF" by Ulrich et al. (2019).

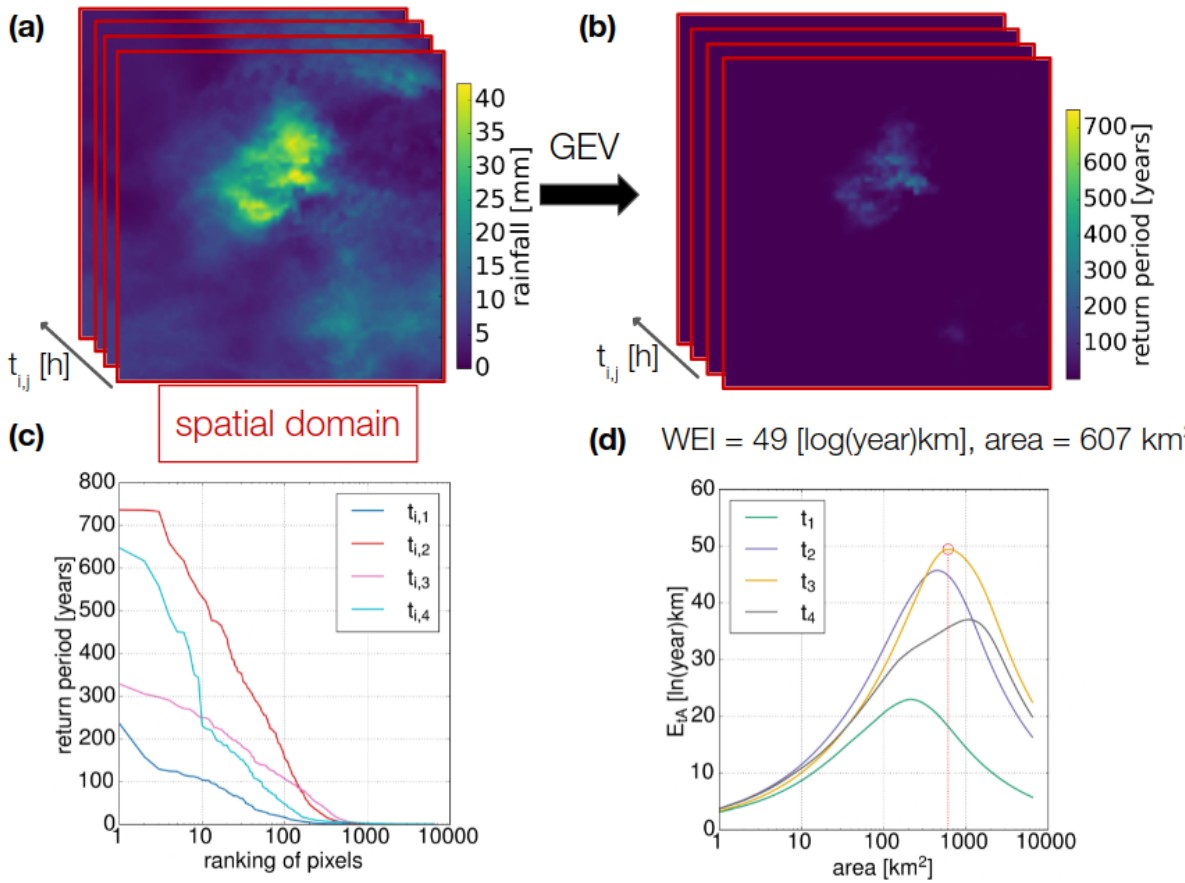

**Figure 1.** Explanation of the *WEI*: (a) Rasters with rainfall intensities at duration $t_i$ for a spatial domain (red) that captures an HPE for each hour ($t_{i,1}$-$t_{i,4}$) of the event. (b) For each pixel and for each timestep ($j = 1, ..., 4$) the return period is calculated. (c) The pixels are sorted by return period in descending order. (d) For each group of the ranked pixels, $E_{tA}$ is calculated. Step (b) to (d) are repeated for all durations $t_i$. In this exemplary case with four timesteps and 4 durations ($i = 1, ..., 4$) this results in 16 $E_{tA}$-curves. The plot (d) shows only the $E_{tA}$-curve with the highest maxima for each duration $t_{1,...,4}$. The maximum of all the resulting curves is the *WEI* (49 [ln(year)km], encircled in red). The maximum was achieved at $t_3$ at a spatial extent of 607 km$^2$.

## 3.2 Calculation of WEI

170    The *WEI* is based on the assumption that an increased extremeness of an event is either due to an increase in intensity or an increase in spatial extent. Hence, the *WEI* is a measure of rarity and spatial extent (Müller and Kaspar, 2014). For a specific duration and a fixed spatial domain, we compute an event's return period for each pixel, and sort the pixels in descending order, based on their return period. The maximum considered return period was set to 1000 years following the example of Müller and Kaspar (2014). Starting with the pixel that has the highest return period, we then successively add one more pixel 175    with the next lower return period. For each set of pixels that results from this incremental process, we compute a measure of





extremeness (Fig. 1). This measure, $E_{tA}$, quantifies the intensity (the average of the return periods) at a specific spatial extent (the number and size of pixels). More specifically, the extremeness $E_{tA}$ for a duration $t$ of a set of $n$ pixels is the product of the mean of the common logarithm of the return periods $p_{t,i}$ and a weighted measure of the area (for which Müller and Kaspar suggested the radius $R$ of a circle whose area $A$ is equal to the pixel group area). The radius $R$ therefore represents a weighting function of the area.

$$E_{tA} = \frac{\sum_{i=1}^{n} ln(p_{t,i})}{n} * \frac{\sqrt{A}}{\sqrt{\pi}} \qquad [ln(year)km] \tag{2}$$

As the pixels are sorted, the average of the return periods is continuously decreasing with each pixel that is added while the area $A$ is continuously increasing. Hence, the extremeness $E_{tA}$ increases as long as cells with high return periods are accumulated. At some point, when more and more pixels with lower return periods are added, the expanding area does not compensate for the decrease of the mean return period, thus leading to a decrease of $E_{tA}$. $E_{tA}$-curves are then calculated for each duration $t$ of interest (Eq. 2) and for each hour (moving window for durations longer than one hour) of the event. The *WEI* is the maximum $E_{tA}$ value found during this procedure. This way the most extreme duration and the spatial extent of an event can be estimated (Fig. 1).

### 3.3 The cross-scale weather extremity index

The calculation of the proposed cross-scale index *xWEI* directly builds on the procedure to compute the *WEI*. We just interpret the $E_{tA}$-curves differently. Each $E_{tA}$-curve displays how the extremeness of an event extends across spatial scales. Hence, such a curve contains more information than just the maximum value (Fig. 2, (a)): the distribution of $E_{tA}$ across scales. The curve informs us whether $E_{tA}$ is high across a larger range of spatial scales (i.e. areas), or whether high values are rather limited to a specific spatial scale. Consequently, the extremeness of an event across spatial scales could be described by the integral of the $E_{tA}$-curve (Fig. 2, (b)). Analogously, we can also integrate $E_{tA}$ across durations in order to measure by how much the extremeness extends across temporal scales. If the $E_{tA}$-curves are represented by a two-dimensional grid along the dimensions area [km$^2$] and duration [h], the $E_{tA}$ curves define a surface that illustrates the extremeness of an event across spatial and temporal scales (Fig. 2, (c)). The surface is derived by interpolation. To ensure that we do not overemphasize long durations in the integral, we used the natural logarithm of the duration. We propose that the volume underneath this surface represents the cross-scale extremeness of an HPE and can be used as a corresponding index which we refer to as *xWEI*. Formally, *xWEI* corresponds to the double integral of $E_{tA}$ over $ln(t)$ and $A$.

$$xWEI = \int_{ln(t)} \int_{A} E_{tA} \, dA \, d(ln(t)) \tag{3}$$

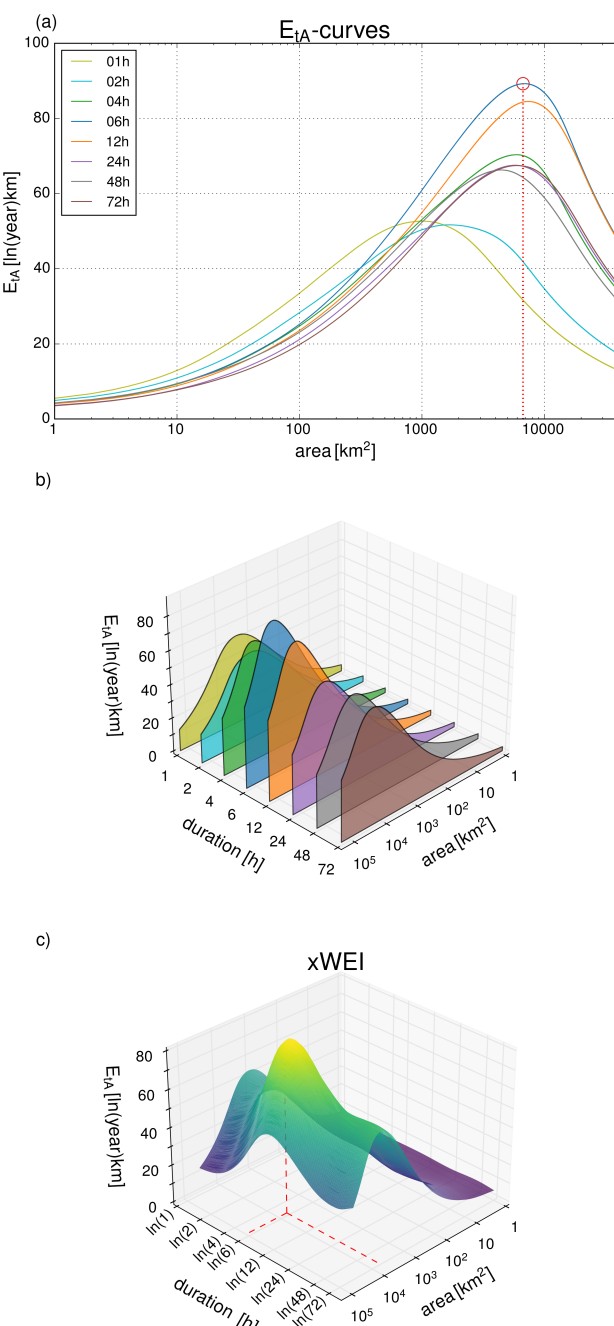

**Figure 2.** (a) $E_{tA}$-curves for different durations, plotted along the spatial dimension (area), with the *WEI* marked by a red circle; (b) the same $E_{tA}$-curves as in (a), but aligned along the temporal dimension (duration); (c) $E_{tA}$-values from (b) interpolated on a regular grid of logarithmic values of durations and area. All plots display the same extreme precipitation event (NW/Jul2014).





## 3.4 Choosing the size of the spatial domain

*WEI* and *xWEI* can be computed for arbitrarily defined spatial domains. The selection of the domain is a subjective decision to
be made by the user, and could have an essential impact on the resulting values of *WEI* and *xWEI*. Müller and Kaspar (2014)
selected the whole country, the Czech Republic, as a spatial domain. Lengfeld et al. (2021a) computed the *WEI* for groups of
contiguous grid cells for which a specific precipitation threshold was exceeded.

In general, WEI and xWEI of different events are comparable only if the size and shape of the spatial domain is the same
across events. The location of the spatial domain can be fixed (e.g. in the case of a country or a river basin), but it could also vary
in space in case we want to compare events that took place in different parts of a larger region (e.g. a large country, continent,
or model domain). The latter applies to our study in which we compare events that occurred in different parts of Germany.
As we are most interested in the cross-scale extremeness of precipitation events in relation to compound inland flooding and
disaster response, we did not compute *WEI* and *xWEI* for the entire spatial RADKLIM domain of 1100 x 900 km. Instead,
we used the centroid of each event selected from the CatRaRE catalog, and then chose, as the spatial domain to compute
*WEI* and *xWEI*, a 200 x 200 km window around that centroid. While this is an arbitrary choice, we considered this size as an
adequate compromise: it is large enough to capture properties that are relevant for the generation of large river floods, but small
enough so that small-scale intense precipitation features (relevant for pluvial and flash floods) are not outweighed by large
scale features. Potential implications of such choices are discussed in section 4.3.2.

## 3.5 Ranking of extreme events

In order to demonstrate the informational value and the behavior of the cross-scale index *xWEI* in comparison to the established
*WEI*, we analyzed the 100 HPEs with the highest *WEI* in the CatRaRE catalog. The event WG/Jul2021, which caused the floods
in western Germany in July 2021, was added to this list although it was not yet included in the catalog, leading to a total of 101
analyzed events. We only used the CatRaRE catalog to select events, but computed both *WEI* and *xWEI* uniformly based on
our above definition of the spatial domain of analysis. Both *WEI* and *xWEI* were than used to rank the events, and to compare
both rankings. Furthermore, we investigated how the choice of the GEV-parameter estimation method (section 3.1) affected
the computation of *WEI* and *xWEI* and hence the ranking results.

## 4 Results and discussion

### 4.1 Effects of GEV parameter estimation method, ranking of events

We calculated the *WEI* and *xWEI* indices for 101 HPEs. Figure 3 shows the corresponding rankings with regard to both indices.
Before we discuss whether and how these rankings depend on the choice of one of the two indices, we would like to briefly
evaluate the sensitivity of the indices on the method to estimate the GEV parameters. The calculation of *WEI* and *xWEI* is based
on calculating the return periods for each duration and each grid cell which is affected by the event. We used three different
methods to derive the return periods (cellwise fitting of GEV distribution, Region of Interest and duration dependent GEV



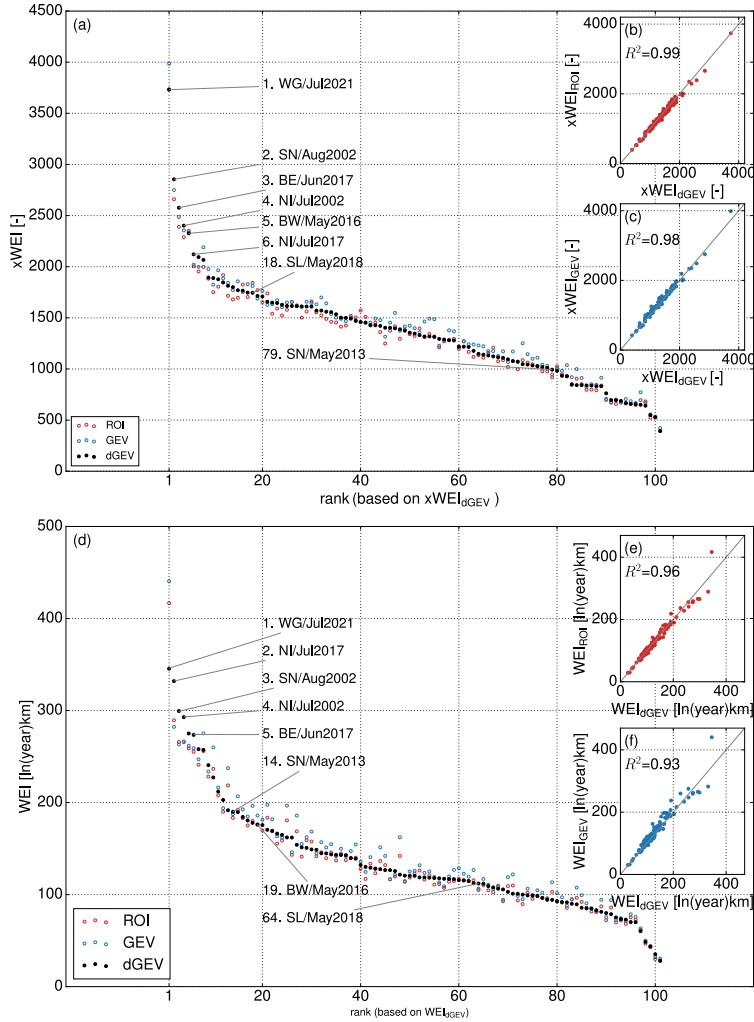

**Figure 3.** (a) Ranking the 101 most extreme precipitation events according to CatRaRE based on the novel *xWEI* index. *xWEI* was calculated with three different methods: cellwise GEV (blue), Region-of-Interest (ROI, red) and duration dependent GEV (dGEV, black), the ranking was carried out based on the values obtained from the dGEV method. (b) comparison of dGEV and ROI method for the calculation of *xWEI*. (c) comparison of dGEV and cellwise GEV. (d) Ranking the 101 most extreme precipitation events according to CatRaRE based on the *WEI*. (e) comparison of dGEV and ROI method for the calculation of *WEI*. (f) comparison of dGEV and cellwise GEV method for the calculation of *WEI*.

distribution). Figure 3 shows the *xWEI* results for each estimation method and each of the 101 evaluated HPEs. Generally,
all three methods lead to very similar results for *WEI* and *xWEI*. The *xWEI* and *WEI* values achieved with the cellwise GEV
method deviate most from the results of the other two methods, while the results of the ROI method and the dGEV are more
similar. This is presumably caused by the fact that the GEV method is less robust, as parameters are estimated separately



for each grid cell and duration. This could affect the ranking, especially for the lower ranks, which is why we discarded this method. The differences between ROI and dGEV are less pronounced. However, we chose the results obtained from the dGEV

method for all of the following analysis. This way, we avoid inconsistencies across durations (such as quantile crossing, see Koutsoyiannis et al., 1998) which is an important feature for the computation of *WEI* and *xWEI* (both indices put return periods from different durations into one context).

With regard to the ranking of events, the WG/Jul2021 event stands out as the most extreme event for both *WEI* and *xWEI*. Furthermore, the events SN/Aug2002, BE/Jun2017, NI/Jul2017 and NI/Jul2002 are ranked among the six most extreme events

for both the *WEI* and *xWEI* index. Interestingly, the NI/Jul2017 (*WEI* rank: 2, *xWEI* rank: 6) event outranks the famous SN/Aug2002 event, that flooded the city of Dresden, (*WEI* rank: 3, *xWEI* rank: 2) when ranked by the *WEI*. Fig. 4 shows four of the highest ranking events with regard to xWEI; The surfaces illustrate the cross-scale extremeness of these events, but they also illustrate, in comparison, the unique level of extremeness of the WG/Jul2021 event.

In this study, however, we are specifically interested in events for which the rank substantially differs between *WEI* and

*xWEI*. The BW/May2016 event, for instance, caused a series of devastating flash floods in south-west Germany, including the notorious flash flood in the village of Braunsbach (Bronstert et al., 2018): this event is ranked at position 19 using the *WEI*, but among the top 5 events based on the *xWEI*. Fig. 5a provides a more systematic representation of how the ranks change subject to the chosen index. The mean absolute deviation between the two rankings is 18 ranks. For 60 events, the ranks based on *WEI* and *xWEI* deviate by 10 ranks or more, for 39 events by 20 ranks or more, for ten events by 40 ranks or more; the maximum

difference is 65 ranks (SN/May2013). To better understand these differences, we selected two case studies: for case study 1, the rank based on *xWEI* is by 47 points lower than the respective *WEI* rank (event SL/May2018, see section 4.2.1); for case study 2, it is by 65 points higher (see section 4.2.2).

Surely, we need to be aware that a small change in the index value could already cause a notable change in rank, specifically beyond the "top ten" ranks where the curves in Fig. 3 are less steep. Hence, the rankings and their comparison, are, to some

extent, sensitive to random effects. Still, Fig. 5(b) confirms that the scatter which we observe in Fig. 5(a) is not just caused by small differences in the index values, but that plotting *xWEI* over *WEI* also exhibits a considerable scatter.

## 4.2    Case studies

### 4.2.1    Case Study 1: SL/May2018

In the night from the 30th of May to the first of June 2018, the small towns of Kleinblittersdorf, Bliesransbach and St. Ingbert

in the federal state of Saarland were hit by a flash flood that also carried a lot of sediment and debris and hence caused essential damage. The event's rank based on *WEI* is 65; based on *xWEI*, it is 18 - the difference in ranks is hence 47. How can we explain such a difference? The plot of the $E_{tA}$-surface (Fig. 6) reveals that this HPE was extreme across temporal scales. For all durations, the maximum $E_{tA}$-values were high (Tab. 2), and even exhibited two local maxima, one around 4 hours and one around 48 hours duration with the maximum $E_{tA}$ at a spatial extent of 5377 km². In total, that leads to a large volume under

the surface spanned by the $E_{tA}$-curves (Fig. 6). The resulting *xWEI* is 1745 [-]. The fact that this event was obviously not

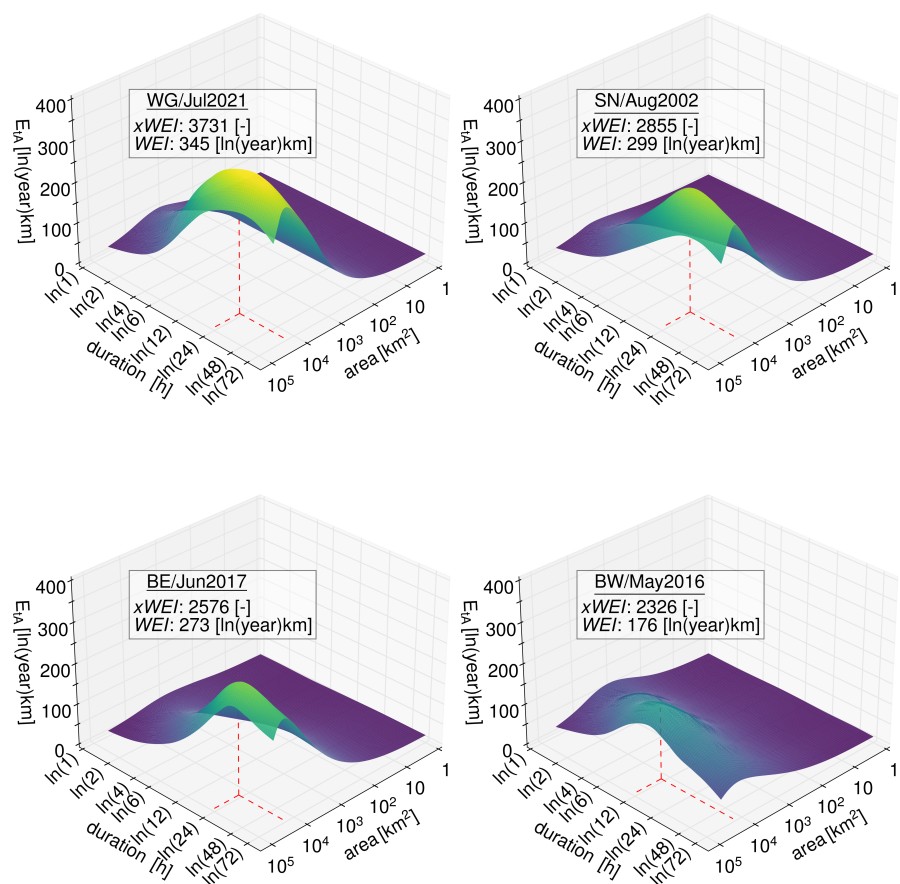

**Figure 4.** Comparison of *xWEI* for four high ranking HPEs. WG/Jul2021 (top left), SN/Aug2002 (top right), BE/Jun2017 (bottom left), BW/May2016 (bottom right). The red lines indicate the spatial and temporal scale for which the event reached its maximum extremeness.

only extreme at a duration around 4 hours is not captured by the *WEI*. Lengfeld et al. (2021a) mention another event in a case study that caused considerable damage in the city of Münster (NW/Jul2014, Fig. 1) which was evaluated with a surprisingly low *WEI* in CatRaRE. Although this event is not included in the original top 101 HPEs in CatRaRE, we re-evaluated this event with *WEI* and *xWEI*. Due to the extremeness on various scales, the *xWEI* would have ranked this event in the top 100 HPE of

CatRaRE (rank 76) and evaluated this event more extreme than the *WEI* (*xWEI*: 1017, *WEI*: 68 [ln(year)km]). Because also our *WEI*-value for this event would have ranked it in the top 100 of CatRaRE (rank 96) we have to consider that our approach regarding the selection of the spatial domain of an event differs from the method chosen CatRaRE (connected cells) which also affects the evaluation of the extremeness of events.

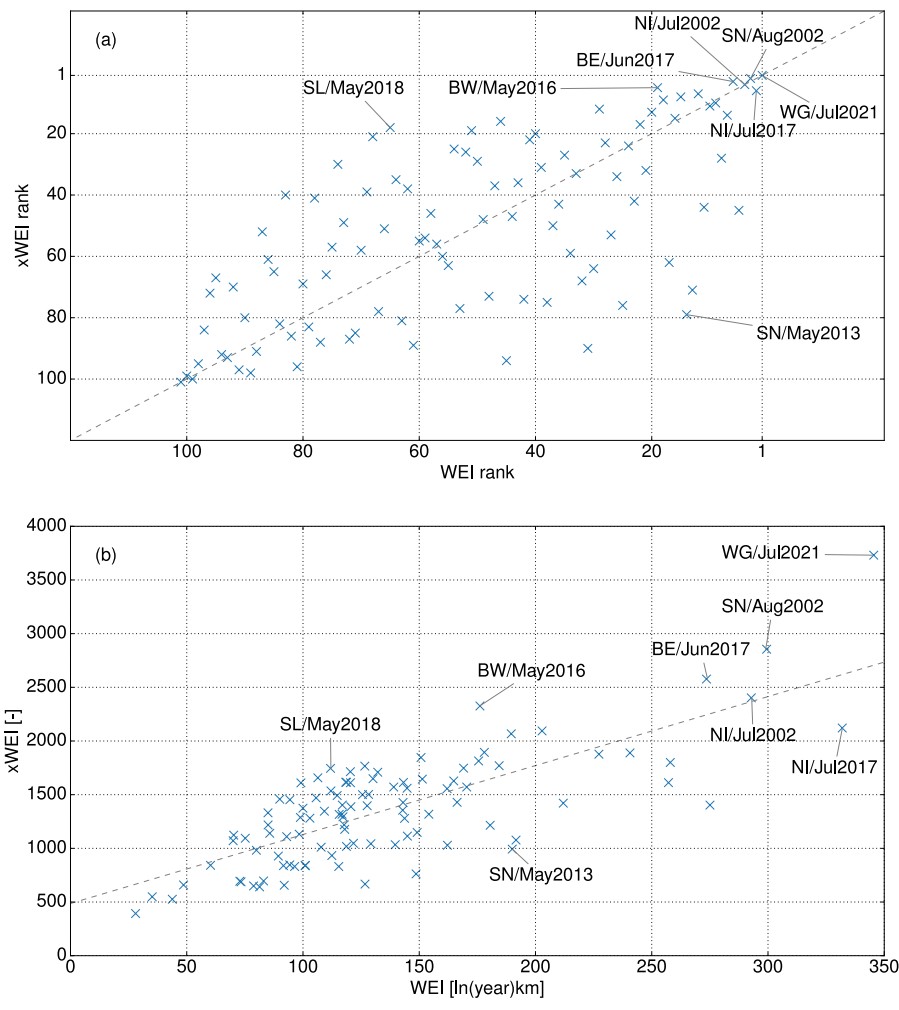

**Figure 5.** (a) *xWEI* ranks plotted over *WEI* ranks. Please note that the values on both axes are decreasing in order to enhance the comparability to subplot (b); (b) *xWEI* values plotted over *WEI* values.

### 4.2.2 Case Study 2: SN/May2013

The second case study shows different cross-scale characteristics as compared to the first. This event lasted from the May 30 to June 2, 2013, with its center in Steinberg/Saxony, and shows extreme precipitation rather at longer durations, with the maximum $E_{tA}$-value observed at the longest analyzed duration of 72 h (Tab. 2 and Fig. 7) and an area of 18,211 km$^2$. The *WEI*-rank for this event is 14 while the *xWEI*-rank is 79. During this event, extreme precipitation on sub-daily timescales is less pronounced. Compared to the SL/May2018 event (Case Study 1), the maximum $E_{tA}$-values for the durations 1 h - 6 h

are relatively low; the *xWEI* for this event is 993 [-]. The maximum $E_{tA}$ at 72 h is represented by *WEI* (190 [ln(year)km]). Based on *WEI*, this event ranks higher than the BW/May2016 event (*WEI*: 176 [ln(year)km]) and the SL/May2018 event (Case





**Table 2.** Maximum $E_{tA}$-values for all considered durations. The *WEI* of this event (the maximum $E_{tA}$ value regarding all durations) is in bold.

| Duration [h] | max. $E_{tA}$ SL/May2018 | max. $E_{tA}$ SN/May2013 |
|:---:|:---:|:---:|
| 01 | 58 | 11 |
| 02 | 80 | 10 |
| 04 | **112** | 18 |
| 06 | **112** | 27 |
| 12 | 90 | 49 |
| 24 | 68 | 79 |
| 48 | 94 | 133 |
| 72 | 72 | **190** |

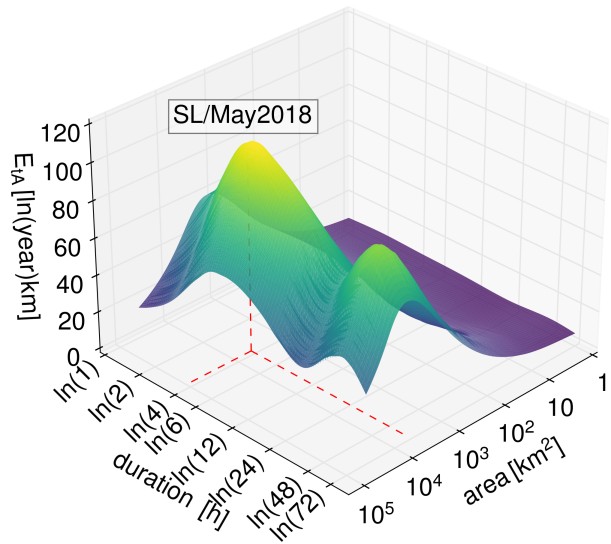

**Figure 6.** SL/May2018. Surface defined by $E_{tA}$-curves. The event shows its maximum extremeness for the durations 4 h and 6 h. Furthermore a second peak at duration 48 h can be observed. The red lines indicate the spatial and temporal scale for which the event reached its maximum extremeness.

Study 1, *WEI*: 112 [ln(year)km]). But the *xWEI* for these events tells a different story: the *xWEI* for the BW/May2016 event (2326 [-]) is 2.3 times higher and the *WEI* for the SL/May2018 event (*WEI*: 1745 [-]) is 1.7 times higher than the *WEI* for the


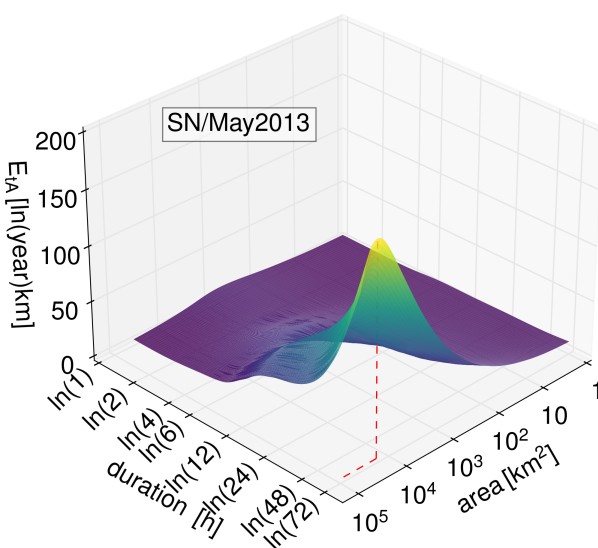

**Figure 7.** SN/May2013. Surface defined by $E_{tA}$-curves. The event is extreme rather at longer durations (maximum at 72 h). The red lines indicate the spatial and temporal scale for which the event reached its maximum extremeness.

SN/May2013 event (*WEI*: 993 [-]). While the SN/May2013 event (Case Study 2) did not show extremeness across temporal

scales, the BW/May2016 event was extreme on all temporal scales with $E_{tA}$-maxima ranging from 60 (1 h duration) to 176 (12 h). Similarly, but less pronounced, this can be observed for the SL/May2018 event.

### 4.3 Required parameter choices for the computation of $E_{tA}$, *WEI*, and *xWEI*

In this section, we will discuss three parameters that affect the computation of $E_{tA}$, *WEI* and *xWEI*, and which need to be set by users who are interested in quantifying the cross-scale extremity of precipitation events. These parameters are: the weight

of the area in the computation of $E_{tA}$, the choice of the spatial domain of analysis, and the choice of duration levels.

#### 4.3.1 Weighting the spatial extent of an event

According to Müller and Kaspar (2014), the *WEI* is the product of a measure of rarity (mean return periods) and a measure of the spatial extent (or area). To avoid that the $E_{tA}$-curves continuously grow with increasing area $A$, the authors suggested to represent the spatial extent by the radius $\frac{\sqrt{A}}{\pi}$ of an imaginary circle with area $A$ (see Eq. 2). While this is an intuitive

and illustrative way to reduce the weight of the area, we need to be aware that the decision to weight the area based on $\frac{\sqrt{A}}{\pi}$ is arbitrary. Weighting the spatial extent differently will change the resulting values of *WEI* and *xWEI*, and hence the corresponding rankings. This arbitrariness should, however, be seen as an opportunity to express preferences with regard to the spatial scale of interest: if we are more interested in local impacts such as flash floods, it might be informative to put less

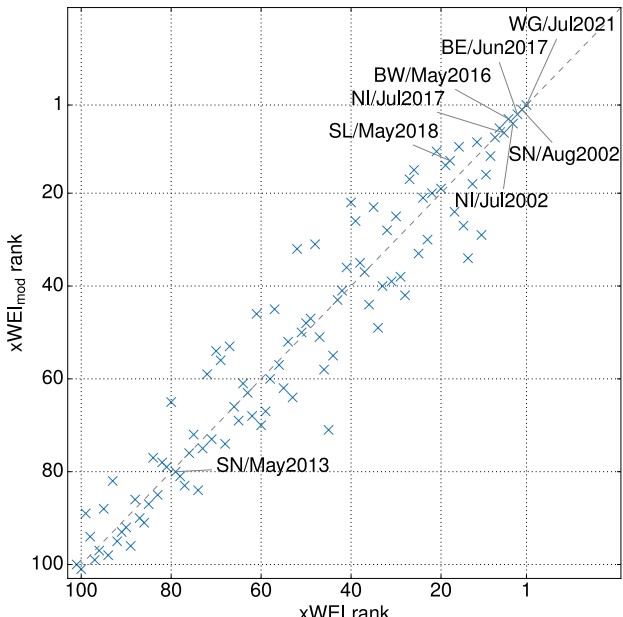

**Figure 8.** *xWEI* compared with each other for two different methods of weighting the area for the calculation of $E_{tA}$. *xWEI* rank describes the weighting proposed by Müller and Kaspar (2014). *xWEI$_{mod}$* was calculated by using the natural logarithm of the area when calculating $E_{tA}$.

weight on the size of the affected area and thus, more weight on cells with high intensity rainfall. This could, for example, be

achieved by replacing $A$ by $ln(A)$ when calculating $E_{tA}$ instead of the radius $R$.

Figure 8 demonstrates how such a choice affects the resulting ranks. While the results are quite similar for the top ten ranks we can observe deviations of 26 ranks for the *xWEI* and up to 59 ranks for the *WEI* (not shown).

### 4.3.2    Setting the spatial domain of analysis

In our analysis, we used a square of 200 x 200 km around the event centroid in order to define the spatial domain for which

the $E_{tA}$-curves as well as *WEI* and *WEI* were computed for each event. Generally, we observed that most high ranking HPEs affected a large spatial domain, and that, for many events, $E_{tA}$ does converge towards zero for very large spatial extents (see e.g. Fig. 4 for the most extreme events). In the previous section 4.3.1, we already discussed the role of weighting the spatial extent of an event when computing $E_{tA}$. Decreasing the weight of the spatial extent, e.g. by using $ln(A)$ instead of $A$, will probably make the falling limbs of the $E_{tA}$-curves steeper, and enhance the convergence of $E_{tA}$ to values of zero with

increasing spatial extent. For many events, though, the value of *xWEI* will grow further if we increase the spatial domain of analysis. The seemingly arbitrary choice of the spatial domain of analysis could be, once more, considered as an opportunity to consider user preferences: while in the present study, we followed the aim of detecting and ranking events across the entire RADKLIM domain, the definition of the spatial domain might be more evident in other contexts. For example, the choice might



be a river basin (see e.g. Gvoždíková et al., 2019), or an administrative unit within which resources for disaster response are

managed. Once fixed, the spatial domain provides a valid frame to compare the cross-scale extremeness of different events up to a maximum spatial scale of interest. In the context of the spatial domain, we also need to be aware that the resulting indices might not represent the full level of extremeness in case the spatial domain of analysis is partly outside the spatial domain for which observations are available. For example, the WG/Jul2021 event extended considerably towards Belgium so that parts of the event were not captured by the radar composite of the German Weather Service.

### 325  4.3.3  Selection and weighting of duration levels

Similar to the above issues of weighting the spatial extent and setting the spatial domain of analysis, the choice of the maximum duration as well as the choice and weighting of duration levels up to this maximum are subject to arbitrariness, or, in other words, to user preferences. This issue is more delicate for the computation of *xWEI* than for the computation of *WEI*: $E_{tA}$ is a function of spatial extent and duration, and as long as the maximum analyzed duration is large enough to detect a local

maximum of $E_{tA}$, the computation of *WEI* is not a problem. The value of *xWEI*, however, will not converge, but grow further with increasing duration levels for as long as $E_{tA}$ does not converge to zero. And even if we chose the maximum duration level large enough for $E_{tA}$ to converge, we need to decide how to integrate $E_{tA}$ over different duration levels in order to compute *xWEI*. Imagine we analyzed all duration levels from 1 to 72 hours with an increment of one hour: in that case, we would have 72 nodes along the duration dimension, of which only 24 would represent extremeness at sub-daily time scales. This imbalance

would overemphasize $E_{tA}$ values at long durations in the resulting estimate of *xWEI*. In the present study, we decided to use 72 hours as the maximum duration, and to integrate $E_{tA}$ along the natural logarithm of duration levels (see section 3.3). That way, the $E_{tA}$-curves for all durations are almost evenly spaced on the two dimensional grid. Still, users might prefer a different maximum duration, and also a different conversion of duration values for the step of integration (which effectively corresponds to putting different weights to different duration levels).

### 340  5  Conclusions and outlook

The *WEI* as suggested by Müller and Kaspar (2014) represents the extremeness of an event at the spatial and temporal scale at which the extremeness reaches its maximum. While such a maximum typically exists, the extremeness of an event can extend across multiple scales - an important property that is not represented by the original *WEI*. The proposed cross-scale index *xWEI* is able to capture cross-scale extremeness in space and time. Accordingly, HPEs might be ranked very differently depending

on which index, *WEI* or *xWEI*, is used. While we do not recommend replacing the original *WEI*, we are confident that the novel *xWEI* can provide valuable complementary information with regard to potential impacts of HPEs. As the computational steps towards the establishment of the underlying $E_{tA}$ -curves are the same for the *WEI* and *xWEI* indices, the added cost of retrieving *xWEI* in top of *WEI* is negligible as compared to the informational benefit.

   In our study, we demonstrated the application and behavior of the *xWEI* index, in comparison to the *WEI* index, for a

set of 101 extreme precipitation events from 2001 to 2021, based on hourly radar-based precipitation composite data of the



German Weather Service (RADKLIM, RADOLAN). We found that the disastrous July 2021 precipitation event in western Germany stood out with regard to both *WEI* and *xWEI*. This similarly applies to other high-ranking events: in our analysis, the events BE/Jun2017, SN/Aug2002 and NI/Jul2002 ranked among the top five for both *WEI* and *xWEI*. Other events were rated as considerably more extreme based on the *xWEI*. Several among these events had become infamous for causing essential damage (e.g. BW/May2016, SL/May2018, and NW/Jul2014 events). As described by Thieken et al. (2022), such damages are often caused by compound inland floods. Generally, we could show that the *xWEI* contains important information about the cross-scale extremeness of HPEs and thus could be used complimentary to the *WEI* which gives information only about the maximum extent and the most extreme duration. The *xWEI* could be a suitable instrument to describe the potential of an HPE to cause impacts, such as floods, and future studies should investigate this potential by systematically linking the *WEI* and *xWEI* indices to observed impacts and damage inventories. To extend the limited scope of this study, future applications should aim at a comprehensive detection and ranking of extreme events from the RADKLIM data set or other similar data sets. The method as described in this study is applicable to any multi-annual gridded time series of precipitation with high resolution in space and time, including radar data as well as regional climate models. Apart from the need to explore the informational value of this novel index in more comprehensive application studies, prospective research should further scrutinize and develop the theoretical foundations of both *WEI* and *xWEI*, and explore the role of subjective choices in the computation of these indices. That particularly applies to

- **the definition of the spatial domain:** in this study, the spatial domain was a window of 200 x 200 $\mathrm{km}$ the center of which varied across the RADKLIM domain. Other window sizes could be used depending on user preferences. In general, the spatial domain should be adjusted to the underlying study objectives: e.g., users could choose the fixed area of a specific river catchment, or an administrative unit that accounts for specific tasks of disaster response.

- **the minimum and maximum duration:** in the present study, the maximum duration was set to 72 hours, according to the standards for extreme value statistics established at the German Weather Service. In analogy to the size of the spatial domain reflecting the maximum spatial scale of interest, the maximum duration level reflects the maximum temporal scale of interest, and could be set accordingly based on user preferences. The minimum duration level was set to one hour, but could be reduced, at least with the RADKLIM data, to 5 minutes. Accounting for sub-hourly durations might shed new light on processes related to pluvial and flash floods, as well as to erosion and landslides.

- **representing the spatial extent for computing $E_{tA}$:** $E_{tA}$ represents the product of rarity and spatial extent. Müller and Kaspar (2014) represented the spatial extent by the radius of a circle with an equivalent area. While this is illustrative, other transformations of the area are conceivable. For example, the natural logarithm of the area would put more emphasis on smaller spatial extents, and cause the $E_{tA}$-curves to drop at a higher rate with increasing spatial extents.

- **weighting spatial extent and duration:** this computational step is specific to *xWEI*; for the computation of *WEI*, we only need to retrieve the maximum value of $E_{tA}$ across duration and spatial extent. For *xWEI*, however, we need to compute the integral of $E_{tA}$ along two dimensions. We noticed that high values of $E_{tA}$ often come along with long



durations and large extents. Using different transformations along these two dimensions could put more emphasis to
smaller scales. Furthermore, it would be more consistent and hence preferable to use the same representation of spatial
extent (e.g. the natural logarithm) for the computation of $E_{tA}$ and for the integration of $E_{tA}$.

Combining a more comprehensive retrieval and comparison of *WEI* and *xWEI* (from RADKLIM or other data) with an
improvement of the theoretical and computational foundations of these indices might open the way to better understand how
the scaling properties of extreme precipitation affect the interaction of processes in compound events, and how they might
affect the processes of disaster risk management across scales.

*Code and data availability.* We published code and data to exemplify the computation of both *WEI* and *xWEI* in the following reposi-
tory: https://doi.org/10.5281/zenodo.6556446. All data used in this study is accessible at the open data repository of the German Weather
Service: the RADKLIM_RW_2017.002 dataset is available at https://opendata.dwd.de/climate_environment/CDC/grids_germany/hourly/
radolan/reproc/2017_002, (Winterrath et al., 2018); the CatRaRE catalog (CatRaRE_2001_2020_W3_Eta_v2021_01) is available at https:
//opendata.dwd.de/climate_environment/CDC/event_catalogues/germany/precipitation/, (Lengfeld et al., 2021b); the precipitation data for
the WG/Jul2021 event was not yet contained in RADKLIM_RW_2017.002 and is available at https://opendata.dwd.de/climate_environment/
CDC/grids_germany/hourly/radolan/historical, (DWD, 2021).

*Author contributions.* PV and MH conceptualized this study. PV developed the software and carried out the analysis; MH contributed to the
analysis. PV prepared the manuscript with contributions of MH.

*Competing interests.* The contact author has declared that neither they nor their co-authors have any competing interests.

*Acknowledgements.* Paul Voit was funded by the Deutsche Forschungsgemeinschaft (grant no. GRK 2043, project number 251036843). We
would like to thank Georgy Ayzel for his support and collaboration in the context of the "ClimXtreme" sub-project CARLOFFF (funded by
the German Ministry of Education and Research (Bundesministerium für Bildung und Forschung, BMBF).



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
