# Peer review of "Quantifying the extremeness of precipitation across scales"

_Natural Hazards and Earth System Sciences, 2022_

## Author Comment (AC3)

**Interactive Discussion: Author Response to Referee #1**

**Quantifying the extremeness of precipitation across scales**

Paul Voit and Maik Heistermann
*NHESS Discussions,* `doi:10.5194/nhess-2022-144`

**RC:** *Reviewer Comment*,     AR: *Author Response*,     ☐ Manuscript text

Dear Referee,

we would like to thank you very much for your willingness to review this paper, and for your swift, positive and constructive response to the manuscript.

Please find our responses to your comments below. These should be considered as preliminary (part of the interactive discussion). The final implementation of changes also depends on another referee report.

Thanks again for your efforts!

Kind regards,
Paul Voit and Maik Heistermann

**RC:** *[...] after reading the article, a general question that remains is the following. The motivation that leads to the definition of WEI is quite clear, we can say -simplifying a lot- that WEI takes the "maximum of maximums" to classify an extreme event. On the other hand, xWEI is somewhat closer to an average WEI across spatio-temporal scales. In this sense, the final xWEI ranking of events might not differ from one obtained using simpler metrics, such as the total precipitation amount integrated over the same spatial scales (e.g. upscale the amounts on coarser grids, by averaging, then integrate). [...] it might be worth showing that the index is more informative than coarser/simpler . In fact, among important users of indices like xWEI there are the providers of climate services. For them, it is quite important that the information delivered conveys an immediate message to the final user. In this sense, a ranking of the events based on e.g. rainfall intensity, rainfall duration, or return period/probability of exceedance may be more appealing. For future research, I suggest you focus also on the comparison against ranking of extreme events based on simpler indicators. It would be interesting to understand the additional information content of xWEI in terms of correlation with registered damages after a catastrophic event, for instance.*

AR: We thank the referee for these ideas, and we agree. As the referee pointed out, these are all valid research questions for future research. While we are hesitant to assume that the *WEI* shows a straightforward relationship to "conventional" event properties such as event intensity or depth, we agree that - once we have established that the *xWEI* conveys important (impact-relevant) properties of extreme precipitation – it could be useful find ways to approximate the *xWEI* by simpler or maybe more intuitive metrics (or combinations of such).

We very much agree with the idea to relate the *xWEI* to observed damages and impacts in order to explore

whether it is able to explain specific damages or damage magnitudes better than other metrics. We had tried to highlight the importance of such an endeavor in ll. 358-360 of the preprint.

**RC:** *Title: You may consider to add "quantifying extremeness of precipitation across scales using the cross-scale weather extremity index xWEI"*

AR: We thank the referee for the suggestion, and we agree that it would be informative to mention that we actually suggest an index to measure cross-scale extremity. However, we would like to keep the acronym xWEI from the title, and we would also like to avoid redundancy in the wording (mentioning extremeness and extremity, as well as across scales and cross-scale). We hence suggest to change the title to "A new index to quantify the extremeness of precipitation across scales".

**RC:** *Abstract. The abstract can be shortened significantly. Try to be short and snappy. For instance, your first 12 lines could be rephrased as (what follows is just an example) "Quantifying the extremeness of a heavy precipitation event is important to classify it. The impact of an event depends on its spatial extent and duration, many indices neglect at least one of these aspects. The weather extremity index (WEI) quantifies the extremeness of an event and identifies the spatial and temporal scale at which the event was most extreme. However, the WEI does not account for the fact that an event can be extreme simultaneously at various spatial and temporal scales. To better understand and detect the compound nature of precipitation events, we suggest to complement the original WEI, and refer to this complement as the "cross-scale weather extremity index" (xWEI). xWEI does not aim to detect the spatio-temporal scale of maximum extremeness, instead it integrates extremeness over relevant scales."*

AR: We thank the referee for the specific suggestions, and we agree that the abstract could and should be shortened. In particular, we will focus the abstract more towards outlining the concept of the cross-scale extremity index instead of providing a comprehensive record of study results. We gladly used your suggestions and shortened the abstract by more than one third, so it becomes:

Quantifying the extremeness of heavy precipitation allows for the comparison of events. Conventional quantitative indices, however, typically neglect the spatial extent or the duration while both are important to understand potential impacts. In 2014, the weather extremity index (*WEI*) was suggested to quantify the extremeness of an event and to identify the spatial and temporal scale at which the event was most extreme. However, the *WEI* does not account for the fact that one event can be extreme at various spatial and temporal scales. To better understand and detect such compound nature of precipitation events, we suggest to complement the original *WEI* by a "cross-scale weather extremity index" (*xWEI*) which integrates extremeness over relevant scales instead of determining its maximum.

Based on a set of 101 extreme precipitation events in Germany, we outline and demonstrate the computation of both *WEI* and *xWEI*. We find that the choice of the index can lead to considerable differences in the assessment of past events, but that the most extreme events are ranked consistently, independently of the index. Even then, the *xWEI* index can reveal cross-scale properties which would otherwise remain hidden. This also applies to the disastrous event from July 2021 which clearly outranks all other analysed events with regard to both *WEI* and *xWEI*.

While demonstrating the added value of *xWEI*, we also identify various methodological challenges along the required computational workflow: these include the parameter estimation for the extreme value distributions, the definition of maximum spatial extent and temporal duration, as well as the weighting of extremeness at different scales. These challenges, however, also represent opportunities to adjust the retrieval of *WEI* and *xWEI* to specific user requirements and application scenarios.

**RC:** *Sec. 2.1. Line 118. When you write that "RADKLIM provides a promising dataset for climatological application", do you mean that it is consistent in time? Can you be a bit more specific*

**AR:** As an operational procedure, RADOLAN is subject to e.g. hard- and software updates over the years or to some rain gauge observations being unavailable in real-time. Therefore, RADKLIM provides a "consistent" reanalysis by using state-of-the-art processing techniques, including new correction algorithms (e.g., for distance- and height-dependent signal reduction) and an enhanced set of of rain gauges for adjustment. The DWD developed RADKLIM with the intent to enable radar-based climatological research and especially heavy rainfall analyses. For these reasons, other authors described RADKLIM as more suitable for climatological studies. We use the term "consistent" instead of "homogeneous" because the issue of heterogeneity due to e.g. changes in radar locations and hardware cannot be entirely resolved by the reanalysis. Furthermore, some parts in the very North, East and South of Germany were only covered for a few years. In general, however, the data coverage over Germany is very good with missing hours of less then 10 % in most areas. Regions of data coverage around 90 % in central Germany are due to exchanges of radar systems. We added additional information and corresponding references in section 2.1.

> The DWD developed RADKLIM with the intent to enable radar-based climatological research and especially heavy rainfall analysis (Kreklow et al. (2019), Winterrath et al. (2018b)). Therefore, the data from 2001 to 2020 was reanalysed by using consistent state-of-the-art algorithms as well as an extended set of rain gauge observations for the adjustment step. It was shown that this procedure minimizes the occurrence of artifacts (Lengfeld et al., 2019) making RADKLIM to a promising dataset for climatological applications (Pöschmann et al., 2019). The resulting dataset is a Germany-wide precipitation field of hourly precipitation sums at an extent of 1100 x 900 km and at a resolution of 1 x 1 km and is available on the DWD open data server (Winterrath et al., 2018a). Parts in the very North, East and South of Germany were only covered for a few years. Otherwise the data coverage is good over Germany with missing hours of less then 10 % in most areas (Lengfeld et al., 2019).

**RC:** *Sec. 3.2. I like your idea of using an example to introduce the WEI. However, I think that: i) you should better describe the initial configuration of your example. At line 174, before "Starting with the pixel..." you may consider adding something like "we will refer to the following example, shown in figure 1, let's consider an event as follows ...""; ii) the definition of the area A is a critical point of the procedure that you discuss again in Sec. 4.3.1, I think that you should let the reader know that you are going to discuss further this point and introduce a reference to Sec 4.3.1 within Sec 3.2. In general, try to create better links between related sections.*

**AR:** We entirely agree, and we will revise the manuscript along the referee's suggestions, including an improved cross-referencing between (sub-)sections. At the same time, we will try to avoid redundancies between the caption of Fig. 1 and the main text.

**RC:** *Sec 3.3. The ratio behind the definition of xWEI is explained in a clear way. I do not completely understand why you need to interpolate the WEI value onto a regular grid (Fig. 2c). It looks to me that one may sum all the areas of the colored curves in Fig 2b and that's it. Since Fig 2b should include all the durations that a user may be interested in, I do not see the risk of overemphasising long durations. Could you add something more on this point?*

**AR:** We very much agree that the referee has a point. We ourselves had the exact same thought: to just sum up or average the area under the individual curves for each duration. This approach is certainly more computationally efficient. Yet, we decided to take the additional step of interpolation for mainly two reasons:

first, the interpolation addresses the effect that the selection of specific duration intervals is somewhat arbitrary: hence we hope that the interpolation allows for a higher level of generalisation, formalisation, and also comparability by acknowledging that there is in fact a surface which we just need to efficiently approximate. The second reason is that we considered, from a visual perspective, the interpolated surface as a more intuitive and coherent representation of the index.

Having said that, we agree that a prospective scrutinization of the xWEI might establish general recommendations on duration choices and weights which allow to approximate the volume under the surface more efficiently, but still consistently.

In order to address the referee's comment, we tried to convey these points in the section 3.3 about the *xWEI* calculation:

> Even though it is computationally more demanding, we chose to interpolate a surface instead of just summing up the integrals of the individual curves (Fig. 2b) to ensure that we seamlessly represent all possible durations, and to avoid overemphasising the arbitrary choice of specific duration levels. Furthermore, we consider the volume under the surface as a more intuitive representation of the index.

**RC:** *Fig 2a. If this is the same as Fig. 1d, then I think you should write it explicitly somewhere in the caption.*

Figure 1d and Figure 2a do not display the same event. The data for Figure 1, although based on an actual event, is simplified and partially altered to improve the clarity of the figure.

**References**

Kreklow, J., Tetzlaff, B., Kuhnt, G., and Burkhard, B.: A rainfall data intercomparison dataset of RADKLIM, RADOLAN, and rain gauge data for Germany, Data, 4, 118, https://doi.org/10.3390/data4030118, publisher: Multidisciplinary Digital Publishing Institute, 2019

Lengfeld, K., Winterrath, T., Junghänel, T., Hafer, M., and Becker, A.: Characteristic spatial extent of hourly and daily precipitation events445 in Germany derived from 16 years of radar data, Meteorologische Zeitschrift, 28, 363–378, https://doi.org//10.1127/metz/2019/0964, publisher: Schweizerbart'sche Verlagsbuchhandlung, 2019

Pöschmann, J. M., Kim, D., Kronenberg, R., and Bernhofer, C.: An analysis of temporal scaling behaviour of extreme rainfall in Germany based on radar precipitation QPE data, Natural Hazards and Earth System Sciences, 21, 1195–1207, https://doi.org/10.5194/nhess-21- 1195-2021, publisher: Copernicus GmbH, 2021

Winterrath, T., Brend, C., Hafer, M., Junghänel, T., Klameth, A., Walawender, E., Weigl, E., and Becker, A.: Erstellung einer radargestützten hochaufgelösten Nieder-schlagsklimatologie für Deutschland zur Auswertung der rezenten Änderungen des Extremverhaltens von Nieder- schlag, https://doi.org/10.17169/refubium-25153, 2018a

Winterrath, T., Brendel, C., Hafer, M., Junghänel, T., Klameth, A., Lengfeld, K., Walawender, E., Weigl, E., and Becker, A.: Gauge-adjusted510 one-hour precipitation sum (RW):, RADKLIM Version 2017.002: Reprocessed gauge-adjusted radar data, one-hour precipitation sums (RW), https://doi.org/10.5676/DWD/RADKLIM_RW_V2017.002, 2018b

---

## Author Comment (AC4)

**Interactive Discussion: Author Response to Referee #2**

**Quantifying the extremeness of precipitation across scales**

Paul Voit and Maik Heistermann

*NHESS Discussions,* `doi:10.5194/nhess-2022-144`
* * *
**RC:** *Reviewer Comment*,    AR: *Author Response*,    ☐ Manuscript text

Dear Referee,

we would like to thank you very much for your willingness to review this paper, and for your swift, positive and constructive response to the manuscript.

Please find our responses to your comments below. These should be considered as preliminary (part of the interactive discussion). The final implementation of changes also depends on another referee report.

Thanks again for your efforts!

Kind regards,
Paul Voit and Maik Heistermann

**RC:** *The authors analyzed 100+1 events with extra high WEI values and determined the xWEI for these events. While I do not suppose that there could be an event with a very high xWEI and yet a WEI so low that it would not belong to the 101 events analyzed, the authors should check this possibility.*

AR: We agree that this possibility exists. The aim of this study, however, is not an exhaustive search for extreme events in the RADKLIM dataset. Instead, we intend to introduce the concept of the *xWEI* and compare *WEI* and *xWEI* for a set of 100+1 events which we selected based on the *WEI* in the CatRaRE catalog. After we have now established that *xWEI* is informative, we will, in the next step, start to systematically and comprehensively extract extreme events from the RADKLIM dataset and compare them with regard to *WEI* and *xWEI*. We had outlined this (computationally expensive) perspective in ll. 360-361 of the preprint. Based on the referee's comment, we suggest to add another sentence thereafter, so the statement will become:

> To extend the limited scope of this study, future applications should aim at a comprehensive detection and ranking of extreme events from the RADKLIM data set or other similar data sets. That way, we might possibly find events with a very high xWEI which were not yet represented in the 100+1 events selected for the present study (based on the CatRaRE catalog).

**RC:** *An important parameter is not only the size of the considered area, but also its shape. The authors should mention this aspect in the article, because the affected area is often elongated in one direction compared to a square.*

AR:   We agree that the shape of the precipitation area is a relevant parameter that deserves further attention for two reasons.

The first reason is more about the relationship between shape and impact. Depending on shape and orientation, the generated runoff might concentrate in a single basin or in multiple neighbouring basins which is likely to have an effect on flood peaks and corresponding impacts. However, this effect also depends on the orientation and shape of affected basins. In general, measures of eccentricity or anisotropy of a rain field could be used as additional metrics in combination with extremity measures such as *WEI* or *xWEI*. To investigate such effects - the interplay of rainfall field attributes and surface properties - should be subject to future research, but is beyond the scope of the present study. We suggest not to discuss this in depth.

The second reason is more at the heart of the present study: a square region of interest might not be able to capture the extent of an event in case of elongated structures. That is, however, a special case of the more general problem in that a region of fixed size might limit the ability to capture the extremeness across scales. In turn, using variable/adjustable regions of interest (variable in size or even shape) makes it difficult to compare (and hence rank) events among each other. One way out of this dilemma could be to normalize *WEI* and *xWEI*, similarly to Gvoždíková et al. (2019). However, such a normalization again introduces further parameters which need to be quantified.

Apart from these specific issues, the actual concept of *xWEI* appears valid and becomes most intriguing if events are to be compared for a region of interest that is not only fixed in size but also in location (see ll. 316 ff. of the preprint): for instance, we can compute the extremeness of events that affected a specific river basin. In that case, it does not play a role whether outside that basin the rainfall process continues to be extreme, as we only want to rank the rainfall that actually affected a well-defined region.

We will emphasize the specific challenge of elongated (frontal) structures in section 4.3.2 in which we discuss setting the spatial domain. In ll. 315-316 of the preprint, we already stated:

> For many events, though, the value of xWEI will grow further if we increase the spatial domain of analysis.

We will add, to this sentence:

> Furthermore, the square shape of the spatial domain might not be an optimal choice to appreciate the extremity of elongated precipitation structures as they e.g. occur along frontal lines [...].

RC:   ***It is also not clear from the paper how the authors dealt with the situation where the core of the event was located at the German border and the 200x200 km square extended beyond the area covered by the data.***

AR:   In lines 322-323 of the preprint, we mentioned the fundamental issue of edge effects:

> In the context of the spatial domain, we also need to be aware that the resulting indices might not represent the full level of extremeness in case the spatial domain of analysis is partly outside the spatial domain for which observations are available. For example, the WG/Jul2021 event extended considerably towards Belgium so that parts of the event were not captured by the radar composite of the German Weather Service.

The problem of the spatial window extending beyond the actual precipitation dataset is not a computational issue in itself (missing values can be treated as zero rainfall). The problem arises in case we have to suspect that substantial parts of the extreme event are hidden behind these missing values, i.e. occured outside the observational domain. As a result, we will underestimate the extremity of events that are close to the edges of the RADKLIM dataset. In the present study, we simply accepted this issue. Instead, one could, for example, discard extreme events of which the centroid is closer than say 100 km to the edge of the data domain. That way, we avoid underestimation, but instead just entirely *miss* important events. Alternatively, we could try to fill missing values around the edges from other data sources, e.g. radar composites or rain gauge data from neighbouring states, or reanalysis model data. Apart from the fact that this would introduce heterogeneity in the data and hence decrease the comparability of events, it would also increase the required resources for data collection and processing. And in the end, there will still be edges.

Altogether, this issue is important to be aware of, but difficult to resolve. As a response to the referee's comment, we propose to enhance the above statement a bit to that it becomes:

> In the context of the spatial domain, we also need to be aware that the resulting indices might not represent the full level of extremeness in case the spatial domain of analysis and the precipitation event extend beyond the spatial domain for which observations are available. For example, the WG/Jul2021 event extended considerably towards Belgium so that parts of the event were not captured by the radar composite of the German Weather Service. The same applies to the SN/Aug2002 event which extended far into the Czech Republic. In fact, we need to acknowledge that the extremeness of events close to the edges of the dataset will, on average, be systematically underestimated. We still decided, for this study, not to discard events that occurred close to the German borders - otherwise, some of the most important events would be entirely missing. Future research, however, could attempt to quantify the systematic errors that are introduced by edge effects.

In section 3.4 we added following part after line 218:

> It is possible that events are not fully captured by this window shape and size but to keep events comparable we decided to stick to a uniform window for all events. Events which are situated close to the state borders of Germany contain more missing rainfall values which adds uncertainty to the evaluation of these events. In some cases the centroid of the event was outside the state borders of Germany. In this case we moved the centroid to the closest pixel with higher rainfall and thereby shifted the spatial domain of the event slightly inwards the borders of Germany. Potential implications of such choices are discussed in section 4.3.

**RC:** *The authors note that the NI/Jul2017 event ranked higher than SN/Aug2002 in the WEI, but offer no explanation (Figure 4 does not include NI/Jul2017). Could the reason be the state-border effect, where SN/Aug2002 significantly affected also the neighboring Czech Republic (Müller et al., 2015)?*

AR: We agree that this could be a reason, based on the discussion of the referee's previous comment. As a response, we would like to extend our previous statement from ll. 245-246:

> Interestingly, the NI/Jul2017 (*WEI* rank: 2, *xWEI* rank: 6) event outranks the famous SN/Aug2002 event that flooded the city of Dresden (*WEI* rank: 3, *sWEI* rank: 2) when ranked by the *WEI*. However, we need to be aware that the SN/Aug2002 event might not have been captured in its full extent by the RADKLIM data as it also affected significant parts of the Czech republic (Müller et al., 2015).

**RC:** *In my opinion, flash- or pluvial floods are mainly related to infiltration excess (line 40) while saturation excess is more typical in case of large-scale fluvial floods (e.g., Rogger et al., 2013).*

AR: We thank the referee for this comment, and we tend to agree. However, we are hesitant to engage, in the context of this paper, specifically in the discussion of runoff generation mechanisms as the relationships between scale, runoff generation and flood processes can be complex (see e.g. Vivoni, 2007). Hence, we suggest to remove the reference to specific runoff generation mechanisms, so that the statement becomes:

> "Short duration rainfall with high intensities is associated to flash- or pluvial floods while persistent precipitation episodes on the daily scale can lead to large-scale fluvial floods."

**RC:** *If the form of the short names of HPEs is your choice, I suggest to replace "NI" by "LS" which seems to be more intuitive in English.*

AR: In order to be as consistent as possible regarding the naming of the events, we appreciate this comment. "NI" will be changed to "LS" in all plots and in the text, as Lower-Saxony is the English name of this German state. This will affect the two events NI/Jul2017 and NI/Jul2002 which will be called LS/Jul2017 and LS/Jul2002 in updated version of the manuscript.

**RC:** *Date formats should be unified, compare e.g. beginnings of both case studies.*

AR: We unified the data formats and hope that all the mentioned dates are now in the same format.

**RC:** *I recommend expanding the beginning of Figure 6 and Figure 7 captions so that they are not just short names of the events. On the other hand, in my opinion, the interpretation of the two pictures does not belong in the captions, its place is in the text.*

AR: We thank the referee for the suggestion. We added the full name corresponding to the text in case study 1 and 2 and removed redundant information about the extremeness of these events to the captions of Figure 6 and Figure 7.

**RC:** *Typos*

AR: The typos have been fixed according to the referees notifications.

**References**

Müller, M., Kašpar, M., Valeriánová, A., Crhová, L., Holtanová, E., GvoÅ¾díková, B., 2015. Novel indices for the comparison of precipitation extremes and floods: an example from the Czech territory. Hydrol. Earth Syst. Sci., 19, 4641–4652.

Gvoždíková, B., Müller, M., and Kašpar, M.: Spatial patterns and time distribution of central European extreme precipitation events between 1961 and 2013, International Journal of Climatology, 39, 3282–3297, https://doi.org//10.1002/joc.6019,2019.

Rogger, M., Viglione, A., Derx, J., Blöschl, G., 2013. Quantifying effects of catchments storage thresholds on step changes in the flood frequency curve.

Vivoni, E. R., Entekhabi, D., Bras, R. L., and Ivanov, V. Y.: Controls on runoff generation and scale-dependence in a distributed hydrologic model, Hydrol. Earth Syst. Sci., 11, 1683–1701, https://doi.org/10.5194/hess-11-1683-2007, 2007.